# The Microbiome of an Outpatient Sports Medicine Clinic During a Global Pandemic: Effects of Implementation of a Microbiome-Specific Cleaning Program

**DOI:** 10.3390/microorganisms13040737

**Published:** 2025-03-25

**Authors:** Greer Russell, Rabia Alegoz, Kelley Hester, Kayla L. Sierzega, Martin J. Szul, Nathaniel Hubert, Timothy Rylander, Sarah Jensen, Mae J. Ciancio, Kristina Martinez-Guryn, Christian C. Evans

**Affiliations:** 1Department of Biomedical Sciences, College of Graduate Studies, Midwestern University, Downers Grove, IL 60515, USA; grussell23@midwestern.edu (G.R.); mcianc@midwestern.edu (M.J.C.); kmarti2@midwestern.edu (K.M.-G.); 2Department of Physical Therapy, College of Health Sciences, Midwestern University, Downers Grove, IL 60515, USA; ralegoz22@midwestern.edu (R.A.); khester99@midwestern.edu (K.H.); kaylasierzega@gmail.com (K.L.S.); 3Department of Microbiology and Immunology, College of Graduate Studies, Midwestern University, Downers Grove, IL 60515, USA; mszul@midwestern.edu; 4Independent Contractor in Bioinformatics Analysis, Mundelein, IL 60060, USA; nathaniel.hubert@gmail.com; 5Impact Physical Therapy, Hinsdale, IL 60521, USA; trylander@impactphysicaltherapy.com (T.R.); sjensen@impactphysicaltherapy.com (S.J.)

**Keywords:** healthcare-associated infection, outpatient rehabilitation clinic, contamination, hand washing, environmental microbiome, COVID-19, cleaning, resistant bacteria, infection, surface microbes

## Abstract

Outpatient healthcare facilities represent potential sources of healthcare-associated infections (HAIs). The purpose of this study was to survey high-contact surfaces in an outpatient physical therapy clinic, characterize the microbiome of those surfaces, and investigate the effects of a microbiome-specific cleaning and hygiene plan. Hand sanitizer containing a fluorescent probe used by patients and staff identified surface contact. High-contact surfaces were analyzed for bacterial DNA and SARS-CoV-2. A microbiome-specific cleaning and hygiene plan was developed based on initial analysis. After the implementation of the revised cleaning regimen, microbial community diversity and predicted metagenome content (PICRUSt) were employed for differential analysis. Patients had greater surface contact than staff. *Ralstonia pickettii* was the dominant species pre-cleaning, comprising 49.76% of the total, and observed on 79.5% of surfaces. The cleaning and hygiene plan significantly increased Shannon diversity, and *R. pickettii* decreased to 4.05% of total bacteria. SARS-CoV-2 was not observed on any surfaces. This study found ecological dominance by a single species in this outpatient clinic, suggesting a potential source of HAIs. However, a microbiome-specific cleaning strategy was successful in diversifying the microbiome and reducing ecological dominance. Additional research is needed to confirm these findings.

## 1. Introduction

As healthcare systems strive to reduce expenditures by decreasing patient length of stay, outpatient healthcare facilities have grown in popularity, allowing patients to rehabilitate at home and in the community [1]. The microbiome of outpatient clinics may contribute to the growing issue of healthcare-associated infections (HAIs) in the United States, particularly due to high levels of patient traffic [2]. By bringing post-hospitalized individuals and members of the broader community into close contact by way of outpatient clinic surfaces, or person-to-person contact, these outpatient healthcare facilities serve as potential sources of HAIs outside of the hospital environment. This may place recently discharged individuals coming from hospitals at a particularly increased risk of infection. For example, a study by Galloway et al., 2016 found the 90-day hospital readmission rate for patients with major disabilities was 34%, and infections were among the most common reasons for readmission [3]. Nosocomial infections, whether acquired in inpatient or outpatient facilities present a significant risk for patients, as increased morbidity and mortality have been observed, as well as heightened hospitalization costs and decreased quality of life [4].

The spread of viral diseases is another concern in HAIs, and the global COVID-19 pandemic has had a lasting impact on infection control in healthcare facilities. Even years after the start of the pandemic, there is still some uncertainty about the role of direct person-to-person and fomite contact [5] and how Severe Acute Respiratory Syndrome Coronavirus 2 (SARS-CoV-2) may be spread in healthcare environments [6]. Moreover, some studies report an association between antibiotic-resistant bacterial infections and the COVID-19 pandemic [7,8,9].

Langford et al., 2023 conducted a meta-analysis of studies examining the incidence of antibiotic-resistant microbes in hospitalized patients comparing pre- and post-COVID-19 pandemic periods and found a non-statistically significant increase in Gram-negative organisms, including *Pseudomonas* and *Acinetobacter* species [8]. Another study by Smith et al., 2023 modeled how the spread of SARS-CoV-2 affected the incidence of antibiotic-resistant infections and found that a rise in SARS-CoV-2 transmission in healthcare facilities increased the spread of these infections in hospital settings [7]. To the best of our knowledge, no studies have examined the impact of COVID-19 on the outpatient clinic microbiome or patterns of surface contamination in outpatient facilities.

While “outpatient healthcare facility” collectively refers to a wide array of services, rehabilitation facilities deserve special consideration for microbial contamination for several reasons. Firstly, physical therapy services require a hands-on approach to care. Contact between healthcare workers and patients is a particularly notable area of microbial spread for both viruses [10] and bacteria [11]. Moreover, healthcare workers can be carriers of pathogens, including antibiotic-resistant strains [12,13]. While the practice of using gloves has been recommended by the World Health Organization [14], studies have shown that gloves only provide protection against hand contamination 87% of the time and do not necessarily protect the patients contacted by gloved workers [15]. While HAIs have been relatively well studied in inpatient facilities such as hospitals [16], few studies have examined the rate of HAIs and transmission patterns in outpatient rehabilitation settings (the authors could find no systematic reviews or studies focused specifically on HAI transmission in outpatient rehabilitation settings). These findings suggest that research should focus on the patterns and sources of HAIs in these facilities.

Studies have shown that environmental contamination of high-touch surfaces is common in the rehabilitative gym setting [11,17], with one study suggesting that of several athletic facility surfaces analyzed, 10% were contaminated with bacteria such as *Staphylococcus aureus* and that surface-to-surface transmission had occurred [17].

Despite advances in understanding the microbiome and patterns of surface contamination in inpatient facilities, the microbial communities in outpatient facilities, and particularly sports medicine physical therapy clinics, are still relatively unknown. Therefore, the purpose of this study was to characterize the microbiome of an outpatient physical therapy clinic, examine surface contact by patients and therapists, and determine whether a microbiome-specific cleaning and hygiene plan could effectively reduce microbial contamination.

## 2. Materials and Methods

### 2.1. Experimental Design, Setting, and Subjects

This observational prospective study was approved by the Midwestern University—Downers Grove IRB. Patients and therapists included in this study agreed to participate by providing their approval following review of a study information sheet. The study site was a private sports medicine/orthopedic clinic in the Midwest USA with a robust clientele (approximately 150 patient visits/week). The clinic was in the suburbs of a major metropolitan center and had no history of healthcare-related infections. At the time of data collection (2021 and 2022), the clinic was implementing standard COVID-19 precautions consistent with the WHO recommendations [14] that consisted of regular handwashing and use of alcohol-based hand sanitizer, thorough weekly deep cleaning, surface-based cleaning after every patient or contact and use of masks and gloves by all staff and masks by patients. Figure 1 shows a timeline of the study with data collection and analysis time points noted relative to the start. A majority of the clientele were seen for orthopedic or sports medicine-related injuries.

### 2.2. Determining Degree of Contact

Prior to the start of this study, the research team met with the clinic owner and staff and identified the most contacted surfaces by both staff and patients based on the staff’s experience working in the clinic (Appendix A shows the specific sites sampled). These surfaces were sampled for patient and staff contact and for bacterial and viral contamination. Based on the methods of Drew et al., the inert fluorescent dye CETEARETH-20 (in the product Glo Germ™ liquid, Glo Germ Co., Moab, UT, USA) was used as an indicator of degree of contact [18]. Staff and patients, on separate days, two weeks apart, were asked use to a 1:1 solution of GermX™ Original Hand Sanitizer and Glo-Germ™ solution made fresh each day. The mixture was placed in specially marked bottles dispersed throughout the clinic for ease of access. Patients and staff, on separate days, were encouraged to reapply the hand sanitizer/Glo-Germ™ mixture throughout the clinic session but were required to apply it at least once at the very beginning of their session. Patient and staff samples were collected two weeks apart to ensure no cross-contamination, with participants in either category applying the hand sanitizer solution only on their specific collection day. Regular hand sanitizer was available for use by the group not being evaluated on a specific day. After covering their hands with sanitizer containing the fluorescent probe CETEARETH-20, they were instructed to go about their normal clinic sessions. At the end of the clinic day (approximately 8 h), surfaces in the clinic were swabbed, and some were visually analyzed using blacklight to inspect for visible levels of fluorescent probes. Cotton-tipped applicators were dipped in dimethyl sulfoxide (DMSO) solution prior to swiping the contacted surface. The applicator tip was broken off into the vile, sealed, and allowed to incubate in 1 mL DMSO solution. Samples were stored protected from light with foil at 4 °C until used.

Degree of contact was determined by measurements of fluorescence for each sample. The sample applicator tip was briefly vortexed in DMSO before the residual solution was loaded in duplicate in 60 µL aliquots onto a black 96-well plate and run on Perkin Elmer’s Enspire© Multimode Plate Reader’s fluorometer. Settings were based on emission and excitation spectra determined using control samples in DMSO. Excitation and emissions wavelengths were set at 403 nm and 434 nm, respectively, and probe concentration was determined from a standard curve of Glo Germ™: DMSO dilutions.

### 2.3. Bacterial DNA Sample Collection, Quantification, and Sequencing

Surface samples were swiped using specially designed synthetic porous swab kits (Norgen Biotek, Inc., Thorold, ON, Canada, Swab Collection and DNA Preservation System™) at the end of a typical clinic day (4 weeks after the collection for the degree of contact measurements). After patients and staff had completed work in the gym on the given day of sample collection, a team of researchers swabbed the clinic surfaces using the Norgen Environmental Sample Kits (Norgen™ Biotek, Inc., Winooski, VT, USA). Approximately 1 ft × 1 ft total surface area for larger surfaces was swabbed using as much lysis buffer solution as was absorbed by the sample kit applicator when dipped. However, surface areas varied, as surface shape depended on equipment type. Samples were placed into separate labeled sterile vials that were preloaded with approximately 1 mL of proprietary lysis buffer from the manufacturer. Sample DNA was isolated and purified using the MasterPure™ Complete DNA and RNA Purification Kit (Epicentre, Inc., Middletown, DE, USA). Bacteria were lysed using zirconium beads in the Mini-Beadbeater™ (Biospec, Bartlesville, OK, USA) for 2 min and then centrifuged at 13,000 RCF for 10 min at 4 °C. Impurities were removed by vortexing in a protein-precipitating buffer. The DNA was precipitated with 100% isopropanol overnight at −20 °C. The following day, DNA was pelleted by centrifugation at 13,000 RCF for 10 min at 4 °C and washed with 70% ethanol. Washed pellets were resuspended in 40 µL of nuclease-free water. DNA quality and purity were checked using the NanoDrop Spectrophotometer™ (Thermo Scientific, Inc., Waltham, MA, USA). DNA samples were partitioned for both total bacterial DNA quantification and community analysis, as described below. 

The Femto DNA Quantification™ kit for total bacterial DNA (ZymoResearch, Inc., Irvine, CA, USA) was used to quantify the amount of DNA based on specific universally conserved regions of DNA [19,20]. The bacterial kit used a universal 16S RNA gene primer set. All samples, standards, and no template controls were loaded in duplicate onto a 96 well-plate and run on the QuantStudio 5 Pro Real-Time PCR System, as previously described [19]. Duplicate CT values for each sample were averaged. 

Community analysis was performed through the application of 16S gene-based amplicon sequencing using the MiSeq Illumina platform (Argonne National Laboratory, Institute for Genomics and Systems Biology, Next Generation Sequencing Core). Sequencing and bioinformatics analysis were performed as previously described [19,20,21]. Primers were based on the 16S rRNA V4-V5 region (F:338F:5′-GTGCCAGCMGCCGCGGTAA-3′ and R:806R:5′-GGACTACHVGGGTWTCTAAT-3′) and contained an Illumina 3′ adapter and 12-base pair barcode. Sequencing output was de-multiplexed and partial overlapping pair-end reads were merged using Illumina Utilities (https://github.com/merenlab/illumina-utils, last accessed 24 November 2024) [22]. Mismatches in overlapping regions of paired-end reads were resolved by selecting the base with the higher Q-score. Merged sequences were retained for downstream analyses if they contained three or fewer mismatches in the overlapping region and if 66% of bases in the first half of each read had an average Q-score of at least 30. Merged reads with any mismatches (>0) were filtered out.

### 2.4. Viral RNA Collection and Detection of SARS-CoV-2

Approximately 4 weeks after collection of the bacterial samples, synthetic nylon swabs (Norgen Inc., Thorold, ON, Canada) were used to swipe the same clinic surfaces for viral RNA. The collection solution contained AVE buffer and lyophilized carrier RNA provided in the kit in 1 mL aliquots. Samples were kept on ice during collection and transport. Extraction was performed using the QIAamp^®^ Viral RNA Mini Kit (QIAGEN, Germantown, MD 20886) for purification of viral RNAs. Samples remained on ice until immediately returned to the laboratory, where they were then stored at −80 °C and not thawed until used. Extraction was performed following the manufacturer’s protocol, including multiple washings using the QIAamp Mini column and double elution of the purified RNA, producing a total of 80 μL of purified RNA solution. Purified RNA was stored at −20 °C until quantification. RNA quantification was performed using the Qubit 3.0™ (ThermoFisher Scientific, San Francisco, CA, USA) and Invitrogen™ RNA HS Assay (ThermoFisher Scientific, San Francisco, CA, USA). Samples were non-diluted and run using the maximum amount of 20 μL from each allotted sample.

A sub-sample of the most contacted surfaces (*N* = 11) was selected for detecting SARS-CoV-2. These surfaces included those identified by the staff as high contact and another 4 chosen by the research team as likely high contact based on results of the contact assay and one surface that tested positive for viral RNA. SARS-CoV-2 detection was performed using the Center for Disease Control and Prevention recommended SARS-CoV RUO positive control (*IDT*), SARS-CoV-2 N1 + N2 Assay Kit (Qiagen, Hilden, Germany), and TaqPath™ 1-Step Multiplex Mastermix (Applied Biosystems™, Waltham, MA, USA) [23]. A 96-well plate was loaded with 12.5 μL of Mastermix, 2.5 μL of N1 + N2 Assay, 25 μL of extracted RNA, and 10 μL of nucleus-free water for a total volume of 50 μL per well. A proprietary combination of four primers and two probes was included in the N1 + N2 Assay based on the CDC design to target N1 and N2 regions of the viral genome. Quantitative PCR was performed on samples run in duplicate on a 96-well qPCR plate in the QuantStudio 5 Pro Real-Time PCR Systems™ (ThermoFisher, Waltham, MA, USA).

### 2.5. Bioinformatics and Statistical Analysis

Analysis of sequencing data was performed based on previously described methods [19] in Illumina-utils. Briefly, quality-filtered reads were partitioned into phylogenetically homogenous units using Minimum Entropy Decomposition (MED) [24] with default parameters. MED iteratively resolves amplicon datasets into high-resolution terminal MED nodes based on Shannon entropy [22]. Taxonomy was assigned to MED nodes using the Genomic Alignment Search Tool (GAST) [25].

Minimum Entropy Decomposition [24] was used to group sequences, and QIIME2 software (QIIME version qiime2-2019.7) [26] was used to assess alpha and beta diversity of surfaces contacted by clinic staff and patients before and after implementation of the cleaning and hygiene plan (hereafter referred to as pre-cleaning and post-cleaning, respectively). Comparison of community compositions was performed using principal coordinate analysis based on Bray–Curtis distances (QIIME version qiime2-2019.7) [26] and plotted with Emperor software 1.0.3 (GitHub, San Francisco, CA, USA, https://biocore.github.io/emperor/, last accessed 12 November 2024) [27].

To assess differential abundance and examine the degree of enrichment or depletion of specific taxa pre- versus post-cleaning, samples were evaluated with Analysis of Compositions of Microbiomes with Bias Correction (ANCOM-BC) with R 2022.12.0 (RStudio Team 2020, Integrated Development for R. RStudio, PBC, Boston, MA, USA, http://www.rstudio.com/, last accessed 22 November 2024). The ANCOM-BC test provided statistical validation while controlling the false discovery rate. Bacterial communities were compared with ADONIS (i.e., PERMANOVA) using Bray–Curtis distances. Analysis for outliers was performed using the ROUT method with Q = 1% (SPSS v27), and total bacteria were checked for normality with the Shapiro–Wilk test using Prism software 9.5.1 (Graphpad, Inc., La Jolla, CA, USA). The amount of residual probe, bacterial DNA, and genus Ralstonia on surfaces was analyzed with Prism 9.5.1 using the Shapiro–Wilk test for normality, ROUT test for outliers (Q-1%), and Wilcoxon matched-pairs test for differences. Correlations were performed using the Spearman test in Prism. Given the potential for non-normally distributed data when examining a naturally occurring microbial community, outliers were identified but not removed. Data were plotted as median ± 25th and 75th percentile with outliers identified as individual data points, and the criteria for significance was *p* < 0.05 for all differences and correlations. 

Functional metabolic gene profiles, pre- and post-cleaning, were predicted utilizing Phylogenetic Investigation of Communities by Reconstruction of Unobserved States (PICRUSt2.4.1) [28]. Analysis was performed using R package ggpicrust2 [29] and used to generate Kyoto Encyclopedia of Genes and Genomes release 102.0 (KEGG; https://www.genome.jp/kegg/, last accessed 22 November 2024) orthologs based on OTC counts from the Greengenes database (v.13.8) [30,31,32].

### 2.6. Clinic Cleaning and Re-Sampling for Bacterial Analysis

After results of the initial viral and bacterial swiping were analyzed (approximately 12 months after the initial swiping), a cleaning and hygiene plan was developed using the data from total surface contamination and sequencing as a guide. An infection control nurse specialist who had experience working with hospital infection outbreaks was consulted in light of the finding of extensive contamination by *R. pickettii*. The plan replaced the general weekly cleaning agent Mr. Clean Multi-Surface Cleaner™, (Procter & Gamble, Cincinnati, OH, USA, active ingredient sodium hydroxide) with a hospital-grade disinfecting cleaning agent, Quaternary Disinfectant Cleaner, (Ecolab, Saint Paul, MN, USA, active ingredient alkyl dimethyl benzyl ammonium chloride) that is considered a broad-spectrum antimicrobial by the manufacturer. Additionally, the daily CaviWipes™ (Metrix^®^, Dubuque, IA, USA, active ingredient isopropanol and ammonium chloride) and Lysol™ Antibacterial Cleaner (Reckitt Benckiser Group^®^, Slough, UK, active ingredient benzalkonium chloride) were replaced with Clorox Bleach Germicidal Cleaner™ (Clorox Healthcare^®^, Oakland, CA, USA, active ingredient sodium hypochlorite) to spot clean after every patient. Other recommendations were to periodically rotate cleaning agents (Quaternary Disinfectant Cleaner and Mr. Clean Multi-Surface Cleaner), continue with hand sanitizers when necessary, but to wash hands with soap and water frequently when time allowed, encourage patients to wash hands or use hand sanitizer, and to continue regular glove use by clinic staff with frequent changing of gloves.

## 3. Results

### 3.1. Clinic Samples and Cleaning

A total of 39 surfaces were initially swiped and analyzed for patient and staff contact, as well as for bacterial DNA (Table 1, Appendix A). Based on the results of the contact assay, 11 of the highest contact surfaces were swiped and analyzed for SARS-CoV-2 (Table 1). After analyzing the results of both the SARS-CoV-2 and pre-cleaning bacterial analysis, the research team consulted an infection control nurse specialist and met with the clinic staff. A cleaning and hygiene plan was developed (Table 2) based on the results of bacterial DNA sequencing [33,34] along with input from the hospital infection control specialist and the clinic staff.

The cleaning and hygiene plan included a hypochlorite-based spot cleaner (Clorox Bleach Germicidal Cleaner™) because a study comparing various common antiseptic cleaners showed that *R. pickettii* was sensitive to this agent [35]. The cleaning plan was implemented over the course of two weeks. Following the microbiome-specific cleaning, 22 of the most contaminated and contacted surfaces were re-swiped and re-analyzed.

### 3.2. Degree of Contact as Measured by Florescent Probe

The clinic map and the inset photographs are available in Appendix A, and show the layout of sampled equipment and examples of fluorescent probes on two surfaces as confirmed by black light inspection. The surfaces examined and the amount of fluorescent probes for each are presented in Table 1. As shown in Figure 2, the results for the degree of contact by patients and staff (Figure 2A) show that of the 39 surfaces analyzed, patients had significantly more contact compared to staff (Patients = 0.00180, 25th/75th percentiles = 0.00060, 0.00390 mg/mL probe, *N* = 38, vs. Staff = 0.000550, 25th/75th percentile = 0.00030, 0.00110 mg/mL probe, *N* = 39, *p* < 0.0001). Surfaces most frequently contacted by patients included the sixth mat table, NuStep™ (Plymouth, MI, USA) recumbent bike handles, and the green resistance cord, whereas the surfaces most contacted by clinic staff included the balance board, Bosu ball, and foot board (Table 1).

### 3.3. Presence of SARS-CoV-2

The initial sampling of all surfaces for viral RNA indicated that only one surface, the handle of the step stool, tested positive for RNA based on the Qubit 3.0 detection system (sample 38; 0.289 ng/µL). Analysis of 11 of the surfaces with the highest contact, including the surface that tested positive for RNA, showed that no samples were positive for SARS-CoV-2 presence, as indicated by no sample measuring over the threshold value (Table 1).

### 3.4. Bacterial Abundance on Surfaces

Total bacterial DNA on surfaces was employed as a proxy for estimating contamination on clinic surfaces. Prior to the implementation of the cleaning plan, bacterial abundance on surfaces was greatest on the third mat table, massage cream tube, and TRX cord handles (Table 1). After cleaning, the most contaminated surfaces were the step stool handles, the forth mat table, and the ultrasound gel bottle (Table 1). After implementation of the cleaning and hygiene plan, and comparing only the samples examined both before and after cleaning, there was no significant change in the mean total bacterial load (Figure 2B, Pre = 0.002950, 25th/75th percentiles = 0.000575, 0.006925, *N* = 22 vs. Post = 0.00030, 25th/75th percentiles = 0.00010, 0.005275, *N* = 22, *p* = 0.4680). Neither the degree of patient nor staff contact showed a significant correlation with total bacteria DNA prior to the cleaning intervention (Patient r = 0.0731, *N* = 21, *p* = 0.6672, Staff r = 0.007443, *N* = 22, *p* = 0.9646). Figure 3 Abundance by genus shown as a histogram of all bacteria for surfaces swiped both pre- and post-cleaning. The key shows the top 45 most abundant taxa resolved as close to the genus as possible, as well as the descriptions of the surfaces sampled. The complete key is shown in the Appendix A. Abbreviations are as follows: Post, post-cleaning; Pre, pre-cleaning; US, ultrasound.

### 3.5. Sequencing Analysis

The 16S amplicon sequencing of highly contacted surfaces revealed the presence of genera commonly associated with human disease (e.g., *Staphylococcus*, *Streptococcus*, and *Pseudomonas*), opportunistic pathogens (e.g., *Acinetobacter*, *Stenotrophomonas*, *Bacillus*, and *Ralstonia*), as well as many other genera not commonly associated with human disease. As shown in Figure 3 and Figure 4, prior to the implementation of the cleaning and hygiene plan, >50% of the total bacteria on surfaces belonged to one genus, *Ralstonia* (58% of the total surface bacteria) with one species, *R. pickettii*, comprising 49.76% of the total surface bacteria. There was a significant difference in the genus *Ralstonia* when comparing relative abundance in MED-nodes per sample in pre- versus post-cleaning samples (Figure 2C, Pre = 0.91012, 25th/75th percentiles = 0.066952, 0.95925, *N* = 21, vs. Post = 0.012014, 25th/75th percentiles = 0.0033605, 0.029865, *N* = 21, *p* = 0.0009).

Additionally, alpha diversity increased post-cleaning compared to pre-cleaning. As shown in Figure 5A, there was no change in richness, but there were increases in evenness (Figure 5B) and overall diversity based on the Shannon Diversity Index (5C). As can be seen in Figure 4, the “other” category, which was comprised of genera that made up <1% of the total pre-cleaning, increased from 10.6% to more than 25% post-cleaning. Pre-intervention, only three of the top 10 bacteria by genera (*Acinetobacter*, *Pseudomonas*, and *Staphylococcus*, comprising 20% of the total) were commonly associated with human skin, whereas post-intervention, four of the five most common bacteria were associated with human skin (*Bacillus*, *Acinetobacter*, *Staphylococcus*, and *Corynebacterium*, comprising 53% of the total) [36].

The cleaning and hygiene intervention resulted in a change in beta diversity (Figure 6). The principal coordinate analysis plot showed that samples clustered differently mainly along the *X*-axis after cleaning compared to pre-cleaning. Analysis of relative abundance (Figure 7) indicated that more than 20 genera were more abundant pre-cleaning, compared to post-cleaning, with *Ralstonia* being the most significant discriminator. Additionally, *Enterobacter*, *Methylobacterium*, *Alloiccoccus*, *Proprionibacterium*, and *Pseudomonas* were markedly more enriched pre- compared to post-cleaning. Post-cleaning, *Bacillus*, *Stenotrophomonas*, *Lactobacillus*, *Amaricoccus*, *Enhydrobacter*, and *Bacteroidales* were significantly enriched and among the most abundant compared to pre-cleaning. Changes in beta diversity are further shown in the Appendix A as a heat map, Appendix A, with the pre- and post-cleaning phylogenetic associations shown in Appendix A.

### 3.6. Associated Metabolic Pathways

The open-source software Phylogenetic Investigation of Communities by Reconstruction of Unobserved States (PICRUSt2) provides insights into the enrichment of pathways associated with observed genera weighted by their taxonomic abundances [29]. The functional metabolic potential associated with pre- and post-cleaning showed numerous processes differentiated on surface environments (Figure 8). Notably, the functional potential of the bacterial communities predicted using PICRUSt2 analysis reflects the observed shift from a community predominated by a single Gram-negative taxonomic group to a more diverse community with an enriched population of Gram-positive bacteria (i.e., *Bacillus* spp.) as highlighted by decreases in pathway abundances associated with lipopolysaccharide biosynthesis and Gram-negative bacterial secretion systems [29]. Additionally, following implementation of the enhanced cleaning and hygiene plan (post-cleaning), we observed a decrease in the predicted abundance of pathways associated with xenobiotics degradation and metabolism, while increases in the predicted pathways for non-ribosomal siderophores, pentose, and glucoronate interconversions and a pathway-associated genetic information repair (i.e., non-homologous end-joining) were observed.

## 4. Discussion

The purpose of this study was two-fold. First, to characterize the clinic surface microbiome in an outpatient sports medicine clinic and determine the relationships between contact and microbial contamination, and second, to determine whether a microbiome-specific cleaning and hygiene plan could reduce the quantity of a specific opportunistic pathogen (i.e., *Ralstonia*). Regarding the first goal of this study, there were several notable findings. While the clinic staff and patients contacted different surfaces, and to different degrees, patients had significantly greater contact as measured by the residual probe. This is not surprising given that patients in a sports medicine clinic are treated on mat tables, use hand-held equipment including resistance bands and weights, and exercise on larger equipment such as ergometers that require extensive hand contact. Additionally, patients rarely used gloves, whereas staff used gloves extensively.

Another interesting and unexpected finding was that while no SARS-CoV-2 was detected prior to the implementation of the cleaning plan, the genus *Ralstonia* dominated the clinic surface microbiome, composing more than 50% of the average surface total microbes by genera. Moreover, more than 90% of the *Ralstonia* present was one species, *R. pickettii*, a cleaning-resistant Gram-negative bacillus known to cause opportunistic infections, especially among immunocompromised hosts [37]. This later finding was surprising in light of studies that show the healthcare-built environment typically hosts a microbiome that is more diverse and reflective of the human skin microbiome [19,38].

Exactly what constitutes the living microbiome of the urban or suburban-built indoor environment is controversial because it is unclear whether current metagenomic techniques actually reflect the true nature of these complex microbial communities [38,39,40]. Reviews by Gilbert and Hartman, 2024, and Bosch et al., 2024 concluded that human hosts, transient and resident animals, and microbes tracked in from the outdoor environment compose most of the indoor microbiome [39,40]. A 2023 analysis of an outpatient rehabilitation clinic, sampled prior to the COVID-19 pandemic, found a similar microbiome, reflective of both human skin and common outdoor surface microbes, with the top six genera being *Staphylococcus* (28%), *Corynebacterium* (22%), *Pseudomonas* (8.3%), *Streptococcus* (7%), *Acinetobacter* (6%), and *Micrococcus* (5.4%) [19]. Although the present study reflects only a single clinic, the high level of *R. pickettii* suggests a significant disruption occurred to offset this balance [41]. The initial samples in the present study were collected between October 2021 and April 2022, during the global COVID-19 pandemic, and may reflect changes in both cleaning and hygiene practices as well as the bacteria present in patients and staff brought into the clinic.

As noted in Figure 7, there were significant changes in the microbiome from pre- to post-cleaning. In addition to a narrower microbial profile (i.e., a small number of bacteria making up the majority of total bacteria), one genus and species was greatly enriched in the community, *R. pickettii*. *Ralstonia* is known to be involved in antibiotic- and cleaning-resistant contamination and HAI. A study by Viana-Ca’rdenas et al. examined an outbreak of *R. pickettii* in hospitalized patients that was reported by the National Institute for Food and Drug Surveillance [34]. This outbreak occurred during March–May 2021, affecting 66 patients from nine hospitals who received intravenous products and thereby developed bloodborne infections. Almost 90% of the affected patients had a COVID-19 diagnosis and most had multiple comorbidities and may have been immunocompromised. This suggests that during the COVID-19 pandemic, patients entering the clinic may have been more exposed to antibiotic- or antiseptic-resistant microbes.

It is known that *R. pickettii* can be resistant to multiple antibiotics, is highly resistant to standard cleaning processes and products, and is a common contaminant of drinking supply water [41], sterile saline, and other medical supplies [42,43]. While *R. pickettii* is considered to have relatively low virulence [44,45], its ability to form biofilms and resist many cleaning and sanitization products and process [46,47] make it a potential pathogen. We can speculate that possibly the excessive cleaning, use of chemical-based hand sanitizers, and over-use of suboptimal and non-hospital-based cleaning products may have selected more resistant microbes such as *R. pickettii*.

There are several possible ways that a significant bloom in a single species, *R. pickettii*, may have occurred in the clinic under study here: first, through a single common source, such as water used at the clinic or from an infected patient or staff member who frequented the clinic; second, due to the use of hand sanitizer or cleaning products that may have been selected for antiseptic or cleaning-resistant microbes such as *R. pickettii*; third, from the use of sanitizers or cleaning products that were themselves contaminated with *R. pickettii*. We applied PICRUSt analysis to our microbiome sequencing results to determine whether the pre-cleaning microbial community was enriched in pathways that may have revealed the source of contamination. The resulting KEGG orthologs library was searched for associations with resistance to cleaning agents or antibiotics, the ability to survive in harsh, dry environments, and spore or biofilm formation. This analysis found that pre-cleaning samples were enriched in pathways involved in the biodegradation and metabolism of xenobiotics, which may have conferred some level of resistance to the specific cleaning products used (e.g., Mr. Clean).

Following the enhanced cleaning plan, we observed an increase in the pathway responsible for non-homologous end-joining (NHEJ). In comparison to homologous recombination, which accurately repairs genome breaks, NHEJ is error-prone and can lead to deleterious mutations in cells requiring its application. In the absence of an intact DNA template that is required for faithful repair by homologous recombination, NHEJ is responsible for the repair of double-stranded breaks which, if left unrepaired, are lethal. The increased presence of organisms capable of NHEJ suggests some level of community adaptation to the stresses induced through the enhanced cleaning plan.

Regarding the potential for the contamination of cleaning products, a scoping review by Lompo et al. examined the relationship between antiseptic and cleaning agent use in healthcare facilities and infection outbreaks [48]. The Lompo et al. study identified several studies in high-income countries that linked bacteria outbreaks to contaminated antiseptic or cleaning products. There were four cases of *R. pickettii* outbreaks, and all four were linked to products that used chlorhexidine as the antibacterial agent. The initial spot-cleaning agent used in this study was CaviWipes, which in addition to alcohol, has the active ingredient didecyldimethylammonium chloride as listed in the manufacturer’s Safety Data Sheet (https://www.life-assist.com/Content/Docs/CaviWipes2_SDS.pdf, last accessed 19 November 2024). We could find no studies linking this antiseptic to specific *R. pickettii* outbreaks or contamination; however, a recent text by Kamph, *Antimicrobial Stewardship*, states that *R. pickettii* has a very high Minimum Inhibitory Concentration for didecyldimethylammonium chloride [49], suggesting that *R. pickettii* may have some resistance to this cleaning agent. This is interesting since both the initial spot cleaning agent and the new general cleaning product introduced as part of the cleaning and hygiene plan included quaternary ammonia compounds closely related to didecyldimethylammonium chloride.

There are a modest number of studies that have specifically examined the microbiome of outpatient clinics and their relevance to pathogen contamination and the spread of HAIs. Dalman et al. swiped surfaces of multiple fitness facilities that are analogous to the common gym area in the current study [17]. Findings from the Dalman et al. study parallelled our findings here in that areas of low and high contamination were similar. While our study demonstrated high bacterial counts on the hand weights for patient contact and the Bosu exercise ball for therapist contact, so too did the Dalman study. Similarly, areas of low contamination were similar such as the hydrocollator tongs, a nonporous and stainless-steel item used by therapists in the clinic. Gontjes et al. examined the patterns of contamination over time to reveal transmission of pathogens in a nursing home rehabilitation gym [11]. They found that patient contact was most closely linked to contamination, similar to our finding here, and that the spread of pathogens, including MRSA and multidrug-resistant bacteria, occurred from patients to surfaces and other patients [11].

While the Gontjes et al. study involved an inpatient facility, the nursing home gym rehabilitation area was very similar to how patients are treated in outpatient physical therapy and sports medicine clinics, suggesting that pathogens may spread similarly. Additionally, Spratt et al., 2019 and Spratt et al., 2014, examined outpatient physical therapy clinic equipment and supplies for contamination and identified tubes of massage cream, ultrasound gel, and ultrasound units as sources of contamination, including antibiotic-resistant strains [50,51].

Another interesting finding in the present study was the effectiveness of a microbiome-specific cleaning and hygiene protocol in reducing the presence of *Ralstonia* but not reducing the overall bacterial load on surfaces. One of the cleaning agents recommended by the infection control specialist was Ecolab^®^ Broad Spectrum Quaternary Disinfectant Cleaning Solution, which contained similar active ingredients (n-Alkyl dimethylethylbenzyl ammonium chlorides and n-Alkyl dimethyl benzyl ammonium chlorides) as the CaviWipes (Didecyldimethylammonium chloride). However, one significant difference between cleaning products was the general weekly cleaner used initially, Mr. Clean^®^, All Purpose Cleaner, had to be diluted with tap water prior to use, whereas the Ecolab^®^ product came pre-diluted in large bottles ready to use. This may have been significant in terms of contamination with *R. pickettii* given this bacteria is known to be present in water supplies and is difficult to kill with the typical antiseptic products used in drinking water processing.

It should be kept in mind that cleaning healthcare facilities involves protecting against any and all pathogens, including viral, fungal, and bacterial agents, not just a single genus of bacteria, even if it is present at a high level. While *R. pickettii* was reduced substantially, this finding does not necessarily mean that the clinic microbiome was less infectious or protected in any way. However, based on the results of sequencing, the post-cleaning microbiome appeared to be normalized in the sense that ecological dominance was reduced, the clinic had a broader representation of different bacterial genera and greater diversity, and was more similar to the expected human skin microbiota that would be typical of an indoor build environment with a high rate of inhabitation.

In retrospect, it is not surprising that none of the surfaces tested for SARS-CoV-2 were positive. A study by Jan et al. specifically examined surface contamination with SARS-CoV-2 in an outpatient healthcare facility by swabbing frequently used surfaces around a radiation oncology department and running samples using rt-PCR analysis similar to our methods [52]. The results of the Jan et al. study showed that of the surfaces sampled, none were positive for SARS-CoV-2. This is consistent with the more recent evidence suggesting that there is a low risk of surface contamination with SARS-CoV-2 [5].

## 5. Limitations

Given this study was conducted during a pandemic, recent changes in cleaning practices (i.e., the institution of periodic deep cleaning regimens and greater vigilance by the staff and patients to wash their hands or use hand sanitizer and use of masks and gloves) likely altered the results compared to prior to the COVID-19 pandemic. Only one clinic was studied here, with a narrow range of clientele and representing only a specific region of the USA, and this facility may not be representative of all outpatient sports medicine/physical therapy facilities. Additionally, the possibility of bias in the methods does exist. For example, the use of a specific agent such as the GloGerm^®^ product or DMSO solvent used during the contact assay, or sterile saline or DNA/RNA preservation collection fluid used in the collection of the DNA and RNA, may have biased the results or contaminated samples. We attempted to control for this by collecting fluorescent probe samples two weeks apart and bacterial and viral DNA samples more than 4 weeks apart and by using the same procedures and products for bacteria DNA collection before and after cleaning. Another possibility is that the processing of the samples and qPCR may have biased or contaminated them, given *R. pickettii*’s ability to resist sterilization procedures. A final limitation is that we focused on *R. pickettii* as an important surface contaminant that needed to be controlled by altering the cleaning and hygiene strategies at the clinic. However, *R. pickettii* is not considered a common pathogen and is not highly virulent. We chose to target this bacteria because it was present at a very high level and because, even though it is not considered highly virulent [41], it is known to cause infections in immunocompromised people [37,45].

## 6. Conclusions

This study screened for the presence of SARS-CoV-2 and examined the microbiome of an outpatient sports medicine clinic during the COVID-19 global pandemic. While no SARS-CoV-2 was detected, the microbiome was dominated by a single genus, *Ralstonia*, with greater than 90% belonging to a single species, *R. pickettii*. A microbiome-specific cleaning and hygiene plan was developed and implemented over a two-week period. While the total level of bacteria was not reduced, there was a significant reduction in *R. pickettii* and indications of a less ecologically dominant microbiome with greater diversity and more similarity to human skin after the implementation of the cleaning intervention. This study suggests that during the COVID-19 pandemic, cleaning or infection-control practices may have exerted selective pressure on the indoor microbiome of this healthcare facility, shifting the microbiome to a narrow and more cleaning-resistant community.

## Figures and Tables

**Figure 1 microorganisms-13-00737-f001:**
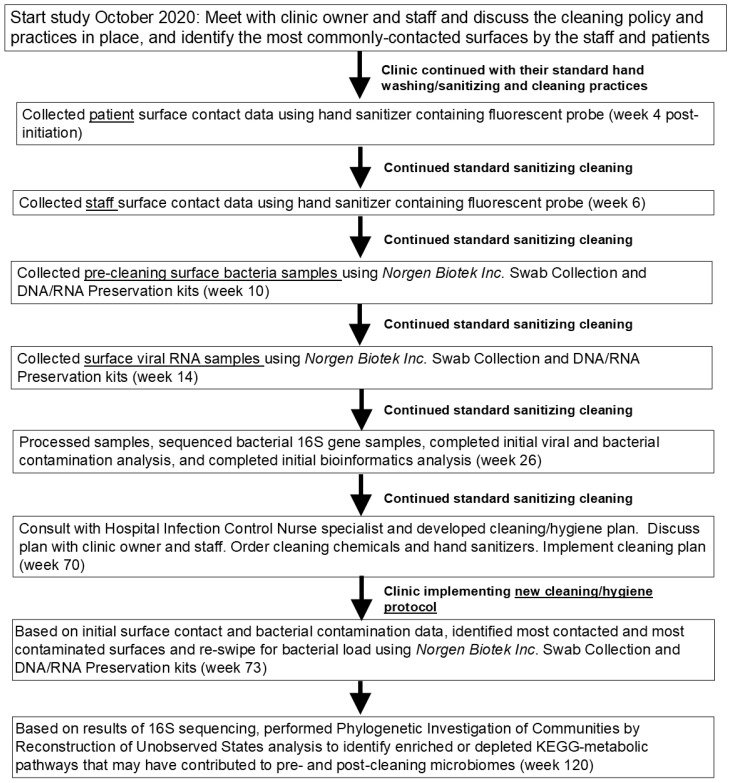
Timeline of study data collection and analysis. Abbreviations are as follows: KEGG, Kyoto Encyclopedia of Genes and Genomes.

**Figure 2 microorganisms-13-00737-f002:**
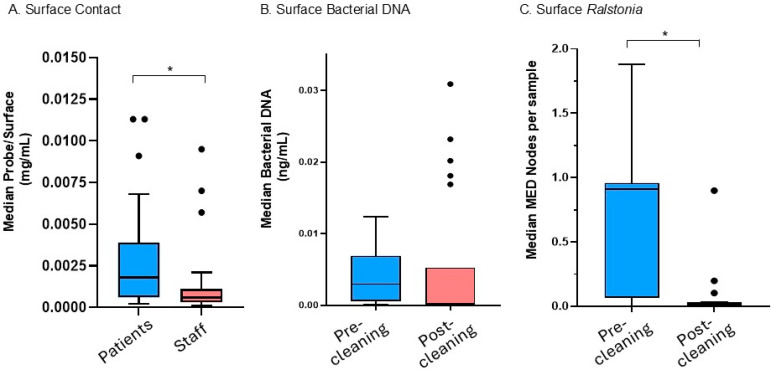
Characterization of surface contact and contamination. (**A**) Patient and staff contact based on residual fluorescent probe. (**B**) Bacterial 16S ribosomal gene DNA detected on surfaces, pre- and post-cleaning. (**C**) *Ralstonia* genus relative abundance (median MED-nodes) per sample based on sequencing analysis, pre- and post-cleaning. Abbreviations and symbols are as follows: * indicates significant difference between groups; mL, milliliters; ng, nanograms; Post, post-cleaning; Pre, pre-cleaning.

**Figure 3 microorganisms-13-00737-f003:**
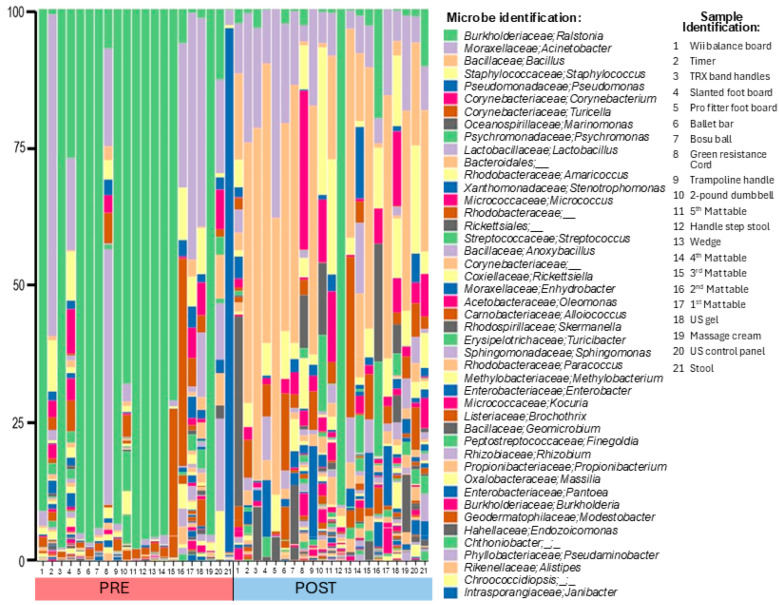
Abundance by genus is shown as a histogram of bacteria pre- and post-cleaning. The key shows the top 45 most abundant taxa resolved as close to genus as possible and surface description. Abbreviations are as follows: Post, post-cleaning; Pre, pre-cleaning; US, ultrasound.

**Figure 4 microorganisms-13-00737-f004:**
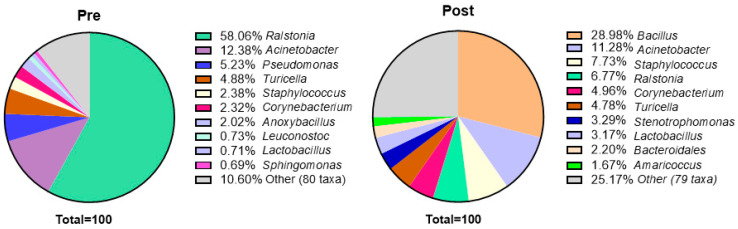
Pie chart showing top 10 genera pre- and post-cleaning. Abbreviations are as follows: Post, post-cleaning; Pre, pre-cleaning.

**Figure 5 microorganisms-13-00737-f005:**
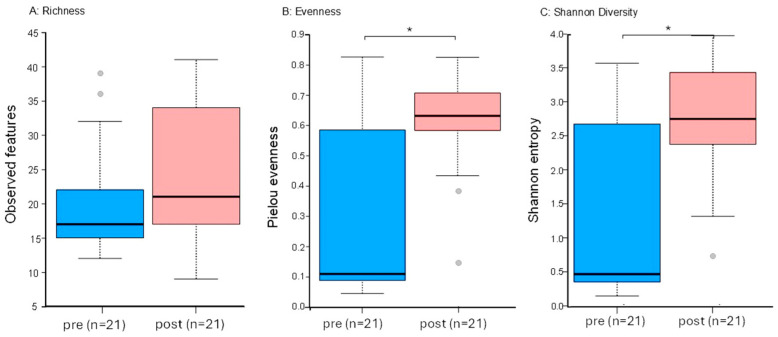
Alpha diversity pre- and post-cleaning based on MED analysis. (**A**) Richness, (**B**) Evenness, (**C**) Shannon Diversity. Abbreviations and symbols are as follows: * indicates significant difference between groups; post, post-cleaning; pre, pre-cleaning.

**Figure 6 microorganisms-13-00737-f006:**
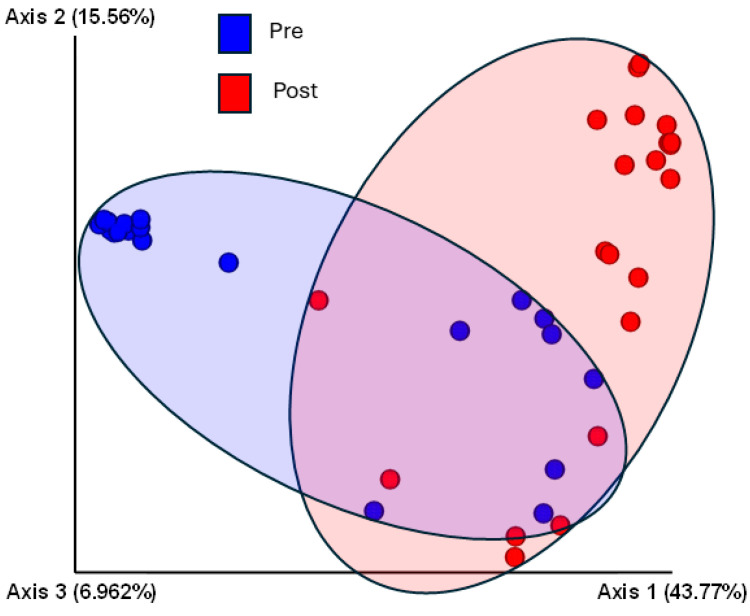
Beta diversity pre- and post-cleaning. The 3rd dimension is coming out of the paper and is not visible. Abbreviations are as follows: Post, post-cleaning; Pre, pre-cleaning.

**Figure 7 microorganisms-13-00737-f007:**
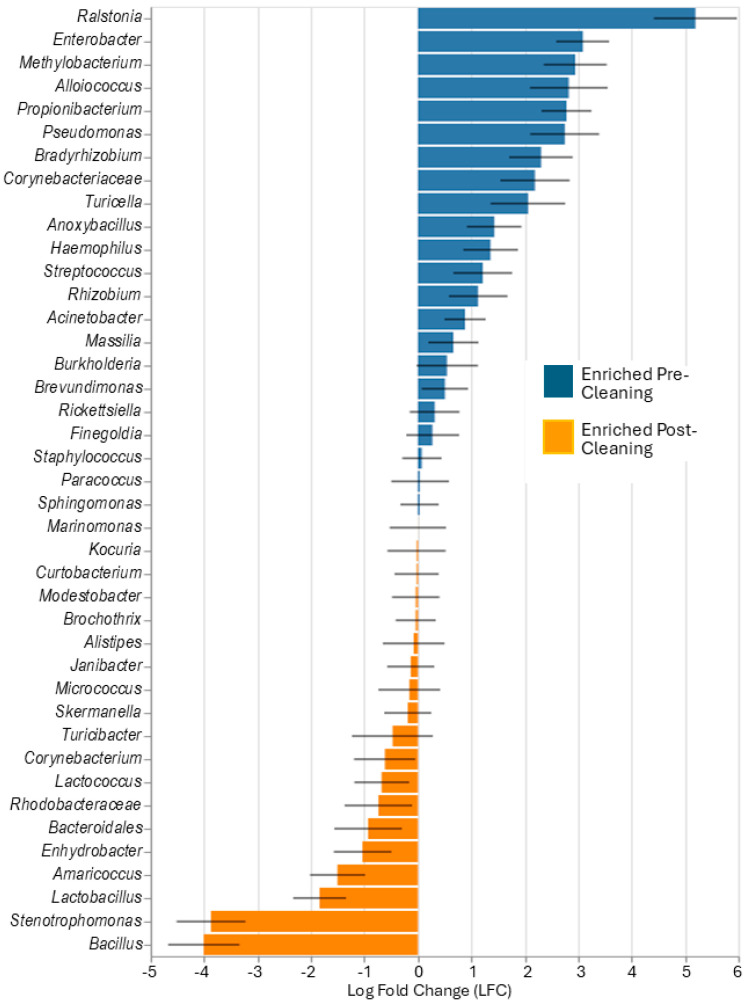
Differential abundance of microbiota based on genus comparing pre- and post-cleaning.

**Figure 8 microorganisms-13-00737-f008:**
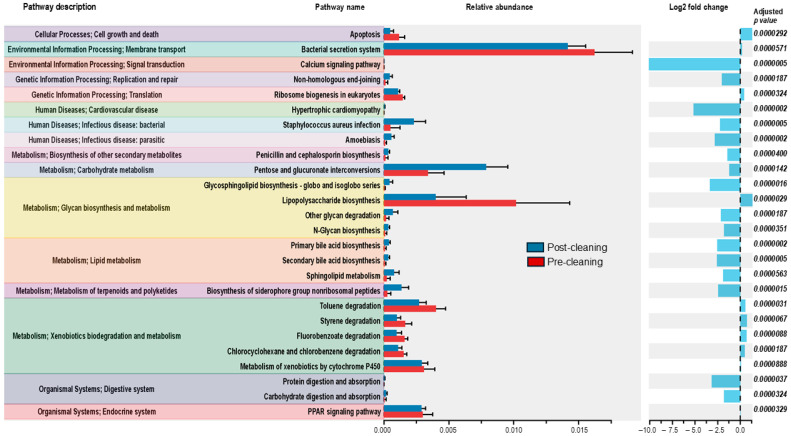
Bar chart showing relative abundance with log fold change and corresponding *p*-values comparing metabolic pathways associated with pre- and post-cleaning surfaces. Abbreviation as follows: PPAR, peroxisome proliferator-activated receptor.

**Table 1 microorganisms-13-00737-t001:** Sample surface characteristics and bacterial contamination.

Site #	Surface Description	Type of Contact	High Contact	Patients Contact (mg/mL)	Staff Contact (mg/mL)	PRE Total Bacteria (ng/mL)	POST Total Bacteria (ng/mL)	Ralstonia (>0.01% OTUs)		SARS-CoV-2
								PRE	POST	
1	Treadmill control panel	hand		0.0011	0.0003	0.0004	NT	+	NT	NA
2	NuStep side handles	hand		0.0113	0.0020	0.0002	NT	+	NT	
3	NuStep arm rest & seat	hand		0.0040	0.0011	0.0003	NT	+	NT	
4	Stationary bike-handle bars	hand		0.0039	0.0004	0.0009	NT	+	NT	
5	Stationary bike-control panel	hand		0.0006	0.0002	0.0007	NT	+	NT	NA
6	Stair stepper side handles	hand		0.0011	0.0003	0.0004	NT	−	NT	
7	Stair stepper control panel	hand		0.0006	0.0002	0.0005	NT	−	NT	
8	Pro fitter foot board	foot	*	0.0039	0.0021	0.0009	0.0014	−	+	
9	Wii balance board	foot	*	0.0012	0.0015	0.0012	0.0001	+	−	
10	PT table stand, top surface	hand		0.0004	0.0004	0.0005	NT	−	NT	
11	Timer	hand	*	0.0004	0.0001	0.0004	0.0001	+	+	
12	Pulley handles	hand		0.0005	0.0001	0.0007	NT	−	NT	
14	Hydrocollator tongs	hand		0.0004	0.0003	0.0002	NT	+	NT	
15	TRX band handles	hand	*	0.0050	0.0008	0.0090	0.0013	+	+	NA
16	Slanted foot board	foot	*	0.0016	0.0057	0.0011	0.0002	+	−	
17	FF balance board	foot		0.0027	0.0095	0.0002	NT	+	NT	
18	Ballet bar	hand	*	0.0061	0.0020	0.0006	0.0169	−	+	
19	Weight bench	other			0.0010	0.0124	0.0181	+	−	NA
20	Bosu ball	hand		0.0028	0.0070	0.0064	0.0002	+	+	
21	Purple exercise mat	other		0.0034	0.0011	0.0002	NT	+	NT	
22	Green resistance cord	hand		0.0091	0.0005	0.0067	0.0012	+	+	NA
23	Trampoline handles	hand		0.0020	0.0005	0.0040	0.0004	+	+	
24	1 lb dumbbell	hand		0.0037	0.0006	0.0004	NT	+	NT	
25	2 lb dumbbell	hand		0.0029	0.0004	0.0076	0.0003	+	+	NA
26	3 lb dumbbell	hand		0.0019	0.0004	0.0003	NT	+	NT	
27	1st mat table	other		0.0020	0.0007	0.0005	0.0004	+	+	NA
28	2nd mat table	other		0.0011	0.0008	0.0002	0.0002	+	+	NA
29	3rd mat table	other		0.0022	0.0009	0.0083	0.0003	+	−	NA
30	4th mat table	other		0.0043	0.0009	0.0001	0.0232	+	+	
31	5th mat table	foot		0.0020	0.0007	0.0004	0.0001	−	−	
32	6th mat table	foot		0.0068	0.0010	0.0056	NT	+	NT	
33	Handle for step stool	hand		0.0024	0.0004	0.0058	0.0309	+	+	NA
34	Wedge	other		0.0017	0.0006	0.0048	0.0002	+	+	
35	Ultrasound gel	other		0.0002	0.0003	0.0022	0.0202	+	+	
36	Massage cream	other	*	0.0006	0.0006	0.0090	0.0001	+	−	NA
37	Massage Gun	hand		0.0013	0.0006	0.0002	NT	+	NT	
38	US control panel & head	hand		0.0011	0.0004	0.0007	0.0001	+	+	
39	Stool	hand		0.0005	0.0003	0.0037	0.0001	−	+	
Total hand, foot and other contacts		Median=		0.00180	0.00060	0.00295	0.0003			
	23 Hand, 6 Foot, 8 Other	25%/75%=		0.0006/0.0039	0.0003/0.0011	0.00057/0.00692	0.0001/0.00527			

Thirty nine surfaces were originally sampled for SARS-CoV-2, bacteria and fluorescent probe contact, and of those, 22 were re-examined for bacterial contamination post-cleaning intervention. Surfaces originally noted as having high levels of bacterial DNA or those determined to be high contact by staff were chosen for re-examination. Degree of contact was estimated by the residual fluorescence on surfaces. Pre and post-cleaning bacterial contamination was determined by qPCR with a universal primer (Femto Total Bacterial Quantification kit™, ZymoResearch). Presence of Ralstonia pickettii was determined from 16S gene sequencing analysis. Comparisons of total bacterial load and Ralstonia genus only included surfaces sampled both before and after the cleaning intervention. Rows with gray background indicate samples collected both before and after cleaning. Abbreviations are as follows: *, identified by staff as high contact; Bact, bacteria; lb., pounds; mg, milligram; mL, milliliters; ng, nanograms; NA, no amplification; NT, not tested; OTU, operational taxonomic unit.

**Table 2 microorganisms-13-00737-t002:** Clinic cleaning and hygiene plan.

Cleaning Strategy	New Products Implemented
Switch from a sodium hydroxide-based weekly floor cleaner (Mr. Clean® All Purpose Cleaner) to multiple hospital-based cleaners. Switch to disinfectant cleaners that did not need to be diluted with tap water.	Ecolab® Broad Spectrum Quaternary Disinfectant Cleaning Solution.
Switch from CaviWipes (Metrix®) and Lysol® Antibacterial Cleaner to a Chlorine-based cleaner for other clinic surfaces.	Clorox™ Bleach Germicidal Cleaner *
Recommend frequent rotation of cleaning agents to ensure microbes do not develop resistance to any one specific agent or compound.	
Switch from reliance primarily on alcohol gel-based hand sanitizers (Purell®) to cleaning hands with soap and water whenever possible.	
Continue regular use of gloves by clinic staff with frequent changing of gloves.	
Encourage patients to wash hands or use hand sanitizer frequently.	

The cleaning and hygiene plan was developed in consultation with a hospital infection control nurse using the results of the initial sequencing study. The new cleaning and hygiene plan was implemented after analyzing the results of the initial data collection, sequencing, and associated analysis and after consulting with the clinic owner and staff (week 70). * *R. pickettii* has been found to be sensitive to chlorine-based disinfectant (Hou et al., 2023).

## Data Availability

The sequencing raw data are available from the National Library of Medicine, National Center for Biotechnology Information, Accession #PRJNA1226985, ID #1226985 at: https://www.ncbi.nlm.nih.gov/bioproject/PRJNA1226985. Last accessed 1 March 2025.

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
