# Peer review of "The Microbiome of an Outpatient Sports Medicine Clinic During a Global Pandemic: Effects of Implementation of a Microbiome-Specific Cleaning Program"

_microorganisms, 2025, doi:10.3390/microorganisms13040737_

Round 1
Reviewer 1 Report
Comments and Suggestions for Authors
Introduction
1. Scientific names of bacteria are not in italics;
2. I suggest dividing larger paragraphs into smaller paragraphs of approximately 5 to 6 lines each paragraph;
Methodology
1. I suggest dividing larger paragraphs into smaller paragraphs of approximately 5 to 6 lines each paragraph;
2. Part of item 2.4 is all in italics
Results
1. I suggest removing figure 1 from the manuscript, I see no need to detail the information in question;
2. I suggest that the authors detail the clinical risk of the bacterial species found in figure 3;
3. The quality of figure 7 does not guarantee the interpretation of the results, I suggest improving it;
4. I suggest improving the interpretation of the results related to figure 8;
5. I suggest improving the quality of figure 9 and also improving the interpretation of the results presented in figure 9; 6. The representation of the P value should be described in italics, please check throughout the document;
Discussion
I suggest improving the formatting of the text, which has different fonts and types.
The quality of the English is good, however, there are some grammatical errors.
Author Response
Response to Microorganisms review of “The Microbiome of an Outpatient Sports Medicine Clinic During a Global Pandemic: Effects of Implementation of a Microbiome-specific Cleaning Program,” Manuscript #3501210
Corresponding author Christian Evans.
3/12/25
Dear Dr. Bong-Soo Kim, Guest Editor, and Microorganisms Review Team:
Thank you and the Microorganisms Reviewers for your thoughtful and detailed review of our submission to the Special Issue of Microorganisms. We are grateful for this chance to revise the manuscript and make improvements. We have addressed the reviewers' recommendations in a point-by-point manner, as shown below, focusing on each review’s comments separately.
In addition, as requested by the journal editor, we uploaded the raw data to a publicly available database: the NIH, NCBI Bioproject database. The data is available under Accession #PRJNA1226985, ID:1226985 at: https://www.ncbi.nlm.nih.gov/bioproject/PRJNA1226985. This was added on page 17, under Data Availability, to the manuscript.
Response to Reviewer #1:
Comment 1: The English could be improved to more clearly express the research.
Response: We have carefully reviewed and revised our use of English language for grammar, typos, or style errors as shown in the “TrackChanges” version of the manuscript attached. These changes were all accepted and incorporated into the final accepted revision that is also attached.
Comment 2: Are the results clearly presented? Marked “can be improved”
Response: We have made specific changes to the results as described in our response to recommendations by both reviewers.
Comment 3: Are the conclusions supported by the results? Marked “can be improved”
Response: We assume that the reviewer is referring to the last sentence of the conclusion where we generalized our results regarding outpatient facilities. We agree with the reviewer and have revised the conclusion to speculate only about changes in the microbiome related to this specific facility. The final sentence now reads: “This study suggests that during the COVID-19 pandemic, cleaning or infection-control practices may have exerted a selective pressure on the indoor microbiome of this healthcare facility, shifting the microbiome to a narrow and more cleaning-resistant community.”
Comment 4: Scientific names of bacteria are not in italics;
Response: The names of bacteria were changed to italics on page 8 in section 3.5. as well as in the first paragraph of the discussion on pages 13 and 14. Additionally, we italicized the names of bacteria in figures 3, 4 and 7.
Comment 5: I suggest dividing larger paragraphs into smaller paragraphs of approximately 5 to 6 lines each paragraph;
Response: We have addressed where possible, given we wanted to retain the continuity of the argument in specific paragraphs. In the Intro (page 2) we divided the second paragraph into two paragraphs (this section changed from 5 to 6 paragraphs) and in the discussion, we divided three paragraphs (changing the discussion from 11 to 14 paragraphs).
Comment 6: Part of item 2.4 is all in italics
Response: This was changed to regular text.
Comment 7: I suggest removing figure 1 from the manuscript, I see no need to detail the information in question;
Response: We have removed figure 1 and added it as a figure in the Data Supplement (DS Fig-1). We replaced figure 1 with a new figure as recommended by reviewer 2. Reviewer 2 recommended a study timeline that showed the data collection time points. This is now referenced in the methods section as Figure 1 (page 3, section 2.1). Please note that we accepted changes to the figures because the trackchanges version because too difficult to work with and read with multiple overlapping figures.
Comment 8: I suggest the authors detail the clinical risk of the bacterial species found in figure 3.
Response: This was added on page 10, section 3.5 directly after figure 3.
Comment 9: The quality of figure 7 does not guarantee the interpretation of the results, I suggest improving it;
Response: We could not find a way to improve the resolution of this figure without enlarging it on the page due to the level of detail and the need for more space with which to view it; therefore, we moved this figure to the Data Supplement section (now Data Supplement Fig-2). We feel this approach addresses the reviewer’s concern as it allows for enlargement of the figure to facilitate the reader’s interpretation. Additionally the information presented in this figure is somewhat redundant with the old figure 8 (the bar graph, now figure 7), given both figures were derived from the sequencing data analysis. The heatmap (DS Fig2) allows readers to qualitatively interpret bacterial diversity in our pre- and post-cleaning samples, whereas the bar graph (new figure 7) presents the overall difference between pre- and post-cleaning surfaces quantitively with the log fold difference and P values indicating significance. The heatmap is included in landscape orientation and enlarged in the Data Supplement, so readers can view the data both ways and interpret the heatmap more effectively.
Comment 10: I suggest improving the interpretation of the results related to figure 8;
Response: The majority of the discussion as currently written is devoted to explaining the results shown in old Figure 8 (now 7), especially as it relates to the enrichment of R. pickettii in the pre-cleaning surface microbiome (Discussion 2nd through sixth paragraphs). To more directly address how the discussion relates to old Figure 8 (new figure 7), we added three new sentences to the beginning of the middle paragraph as shown here: “As noted in Figure 7, there were significant changes in the microbiome from pre- to post-cleaning. In addition to a narrower microbial profile (i.e., a small number of bacteria making up the majority of total bacteria), one genus and species was greatly enriched in the community, R. pickettii. Ralstonia is known to be involved in antibiotic and cleaning resistant contamination and HAIs…”
Comment 11: I suggest improving the quality of figure 9 and also improving the interpretation of the results presented in figure 9; 6. The representation of the P value should be described in italics, please check throughout the document;
Response: We were able to reformat old Figure 9 (now Figure 8) to shrink the empty space, so that the actual bar graph portion could be enlarged and so the KEGG pathways are easier to read. We also italicized the P-values. To better describe the findings in this figure, we added to the 3rd and 4th sentences of the results (Section 3.6, top paragraph). In addition, we currently include two paragraphs of discussion (the 2nd and 3rd paragraphs on page 16) related to interpreting the results of this analysis.
Comment 12: I suggest improving the formatting of the text, which has different fonts and types.
Response: We have reviewed the text and changed all fonts to match.
Comment 13: The quality of the English is good, however, there are some grammatical errors.
Response: As mentioned in the response above, we revised minor English language grammar, typos, or style errors as shown in the “TrackChanges” version of the manuscript attached.
Response to Reviewer #2:
Comment 1: Are the results clearly presented? Marked as “Must be improved”
Response: We believe that we have improved the results and more explicitly described the data and findings in the figures. We kindly ask the reviewer to see our responses to Review #1’s comment #8, 9, 10 and 11 above.
Comment: I have minor comments, although due to the need for improvement of graphic quality, I have recommended major revisions to allow for enough time.
Comment 2: Please adjust paper formatting according to journal requirements.
Response: We believe these adjustments were made and include reformatting figures without legends and including the legend in the text, using the standard journal font and size, using standard journal section headings, and other adjustments.
Comment 3: The first sentence of the introduction needs refinement.
Response: We agree with the reviewer and have made the following change to this sentence: “As healthcare systems strive to reduce expenditures by decreasing patient length of stay, outpatient healthcare facilities have grown in popularity, allowing patients to rehabilitate at home and in the community [McDermott et al 2017]..”
Comment 4: Sections 2.2, 2.3: please describe how were surfaces selected for sampling.
Response: We agree with the reviewer and added two sentences on page 4, in section 2.2 to describe how we selected surfaces for patient and staff contact and bacterial and viral contamination.
Comment 5: Sections 2.2, 2.3, 2.6: it would be useful if the authors presented this procedure in a graphic manner to make it more comprehensible.
Response: We agree and have replaced old Figure #1 (clinic map) with a new figure showing the study timeline, including when surface contact data, surface bacteria contamination, and surface viral contamination data were collected. We moved old Figure 1 to the data supplement.
Comment 6: Section 2.4: Please correct formatting.
Response: This was corrected.
Comment 7: Table 2: in the table legend, please clarify when these practices were applied.
Response: This was corrected by adding the following sentence to the legend of table 2: “The new cleaning and hygiene plan was implemented after analyzing the results of the initial data collection, sequencing, and associated analysis and after consulting with the clinic owner and staff (week 70).”
Comment 8: Figure 3: microbe identification is not very clear.
Response: We reformatted the legend in Figure 3, removing the complete taxonomic hierarchy names of bacteria and limiting it to just the family and genus or the lowest taxonomic level resolved, and increasing the font size and clarity. We feel it is now an acceptable level of resolution.
Comment 9: Figure 6 is probably not necessary.
Response: We respectfully disagree with the reviewer. Figure 6 is the only place in the manuscript where we show the beta diversity, an important indicator that a significant shift occurred In the surface microbiome from pre- to post-cleaning. This figure clearly shows a large shift, based on the change along the X-axis. We ask that this figure be retained.
Comment 10: Figure 7 is not very clear. Please consider revising otherwise omit.
Response: We have omitted this figure and moved it to the Data Supplement where it can be enlarged.
Thank you for your careful review and for this opportunity to revise and improve our manuscript. We look forward to your decision.
Best, Chris
Christian C. Evans, PT, PhD
Professor, Physical Therapy
Midwestern University
555 31st Street, AH 340E
Downers Grove, IL 60515
(630) 515-7249

Reviewer 2 Report
Comments and Suggestions for Authors
This is a well written and well performed study, evaluating the level of surface viral & bacterial contamination in an outpatient sports medicine department. The authors provide a detailed description of a complex methodology and have used several technologies to achieve their study aims. The description of their findings is detailed and corresponds to the study aims and methods, although several graphical depictions need improvement. The discussion is complete, with acknowledgement of strengths and weaknesses, as well as comparisons with existing literature. I have minor comments, although due to the need for improvement of graphic quality, I have recommended major revisions to allow for enough time.
Please adjust paper formatting according to journal requirements.
The first sentence of the introduction needs refinement.
Sections 2.2, 2.3: please describe, how were surfaces selected for sampling.
Sections 2.2, 2.3, 2.6: it would be useful if the authors presented this procedure in a graphic manner to make it more comprehensible.
Section 2.4: Please correct formatting.
Table 2: in the table legend, please clarify when these practices were applied.
Figure 3: microbe identification is not very clear.
Figure 6 is probably not necessary.
Figure 7 is not very clear. Please consider revising otherwise omit.
Author Response

(The authors gave the same response as above.)

Round 2
Reviewer 2 Report
Comments and Suggestions for Authors
All comments sufficiently addressed. Thank you